# Convex Calibrated Surrogates for Low-Rank Loss Matrices with Applications to Subset Ranking Losses

**Harish G. Ramaswamy**
Computer Science & Automation
Indian Institute of Science
harish_gurup@csa.iisc.ernet.in

**Shivani Agarwal**
Computer Science & Automation
Indian Institute of Science
shivani@csa.iisc.ernet.in

**Ambuj Tewari**
Statistics and EECS
University of Michigan
tewaria@umich.edu

## Abstract

The design of convex, calibrated surrogate losses, whose minimization entails consistency with respect to a desired target loss, is an important concept to have emerged in the theory of machine learning in recent years. We give an explicit construction of a convex least-squares type surrogate loss that can be designed to be calibrated for any multiclass learning problem for which the target loss matrix has a low-rank structure; the surrogate loss operates on a surrogate target space of dimension at most the rank of the target loss. We use this result to design convex calibrated surrogates for a variety of subset ranking problems, with target losses including the precision@$q$, expected rank utility, mean average precision, and pairwise disagreement.

## 1   Introduction

There has been much interest in recent years in understanding consistency properties of learning algorithms – particularly algorithms that minimize a surrogate loss – for a variety of finite-output learning problems, including binary classification, multiclass classification, multi-label classification, subset ranking, and others [1–17]. For algorithms minimizing a surrogate loss, the question of consistency reduces to the question of calibration of the surrogate loss with respect to the target loss of interest [5–7, 16]; in general, one is interested in convex surrogates that can be minimized efficiently. In particular, the existence (and lack thereof) of convex calibrated surrogates for various subset ranking problems, with target losses including for example the discounted cumulative gain (DCG), mean average precision (MAP), mean reciprocal rank (MRR), and pairwise disagreement (PD), has received significant attention recently [9, 11–13, 15–17].

In this paper, we develop a general result which allows us to give an explicit convex, calibrated surrogate defined on a low-dimensional surrogate space for any finite-output learning problem for which the loss matrix has low rank. Recently, Ramaswamy and Agarwal [16] showed the existence of such surrogates, but their result involved an unwieldy surrogate space, and moreover did not give an explicit, usable construction for the mapping needed to transform predictions in the surrogate space back to the original prediction space. Working in the same general setting as theirs, we give an explicit construction that leads to a simple least-squares type surrogate. We then apply this result to obtain several new results related to subset ranking. Specifically, we first obtain calibrated, score-based surrogates for the Precision@$q$ loss, which includes the winner-take-all (WTA) loss as a special case, and the expected rank utility (ERU) loss; to the best of our knowledge, consistency with respect to these losses has not been studied previously in the literature. When there are $r$ documents to be ranked for each query, the score-based surrogates operate on an $r$-dimensional surrogate space. We then turn to the MAP and PD losses, which are both widely used in practice, and for which it has been shown that no convex score-based surrogate can be calibrated for all probability distributions [11, 15, 16]. For the PD loss, Duchi et al. [11] gave certain low-noise conditions on the probability distribution under which a convex, calibrated score-based surrogate could be designed;

we are unaware of such a result for the MAP loss. A straightforward application of our low-rank result to these losses yields convex calibrated surrogates defined on $O(r^2)$-dimensional surrogate spaces, but in both cases, the mapping needed to transform back to predictions in the original space involves solving a computationally hard problem. Inspired by these surrogates, we then give a convex score-based surrogate with an efficient mapping that is calibrated with respect to MAP under certain conditions on the probability distribution; this is the first such result for the MAP loss that we are aware of. We also give a family of convex score-based surrogates calibrated with the PD loss under certain noise conditions, generalizing the surrogate and conditions of Duchi et al. [11]. Finally, we give an efficient mapping for the $O(r^2)$-dimensional surrogate for the PD loss, and show that this leads to a convex surrogate calibrated with the PD loss under a more general condition, i.e. over a larger set of probability distributions, than those associated with the score-based surrogates.

**Paper outline.** We start with some preliminaries and background in Section 2. Section 3 gives our primary result, namely an explicit convex surrogate calibrated for low-rank loss matrices, defined on a surrogate space of dimension at most the rank of the matrix. Sections 4–7 then give applications of this result to the Precision@q, ERU, MAP, and PD losses, respectively. All proofs not included in the main text can be found in the appendix.

## 2 Preliminaries and Background

**Setup.** We work in the same general setting as that of Ramaswamy and Agarwal [16]. There is an instance space $\mathcal{X}$, a finite set of class labels $\mathcal{Y} = [n] = \{1, \ldots, n\}$, and a finite set of target labels (possible predictions) $\mathcal{T} = [k] = \{1, \ldots, k\}$. Given training examples $(X_1, Y_1), \ldots, (X_m, Y_m)$ drawn i.i.d. from a distribution $D$ on $\mathcal{X} \times \mathcal{Y}$, the goal is to learn a prediction model $h : \mathcal{X} \rightarrow \mathcal{T}$. Often, $\mathcal{T} = \mathcal{Y}$, but this is not always the case (for example, in the subset ranking problems we consider, the labels in $\mathcal{Y}$ are typically relevance vectors or preference graphs over a set of $r$ documents, while the target labels in $\mathcal{T}$ are permutations over the $r$ documents). The performance of a prediction model $h : \mathcal{X} \rightarrow \mathcal{T}$ is measured via a *loss function* $\ell : \mathcal{Y} \times \mathcal{T} \rightarrow \mathbb{R}_+$ (where $\mathbb{R}_+ = [0, \infty)$); here $\ell(y, t)$ denotes the loss incurred on predicting $t \in \mathcal{T}$ when the label is $y \in \mathcal{Y}$. Specifically, the goal is to learn a model $h$ with low expected loss or $\ell$-error $\mathrm{er}_D^\ell[h] = \mathbf{E}_{(X,Y)\sim D}[\ell(Y, h(X))]$; ideally, one wants the $\ell$-error of the learned model to be close to the optimal $\ell$-error $\mathrm{er}_D^{\ell,*} = \inf_{h:\mathcal{X}\rightarrow\mathcal{T}} \mathrm{er}_D^\ell[h]$. An algorithm which when given a random training sample as above produces a (random) model $h_m : \mathcal{X} \rightarrow \mathcal{T}$ is said to be *consistent* w.r.t. $\ell$ if the $\ell$-error of the learned model $h_m$ converges in probability to the optimal: $\mathrm{er}_D^\ell[h_m] \xrightarrow{\mathrm{P}} \mathrm{er}_D^{\ell,*}$.[1]

Typically, minimizing the discrete $\ell$-error directly is computationally difficult; therefore one uses instead a *surrogate loss function* $\psi : \mathcal{Y} \times \mathbb{R}^d \rightarrow \bar{\mathbb{R}}_+$ (where $\bar{\mathbb{R}}_+ = [0, \infty]$), defined on the continuous surrogate target space $\mathbb{R}^d$ for some $d \in \mathbb{Z}_+$ instead of the discrete target space $\mathcal{T}$, and learns a model $\mathbf{f} : \mathcal{X} \rightarrow \mathbb{R}^d$ by minimizing (approximately, based on the training sample) the $\psi$-error $\mathrm{er}_D^\psi[\mathbf{f}] = \mathbf{E}_{(X,Y)\sim D}[\psi(Y, \mathbf{f}(X))]$. Predictions on new instances $x \in \mathcal{X}$ are then made by applying the learned model $\mathbf{f}$ and mapping back to predictions in the target space $\mathcal{T}$ via some mapping $\mathrm{pred} : \mathbb{R}^d \rightarrow \mathcal{T}$, giving $h(x) = \mathrm{pred}(\mathbf{f}(x))$. Under suitable conditions, algorithms that approximately minimize the $\psi$-error based on a training sample are known to be consistent with respect to $\psi$, i.e. to converge in probability to the optimal $\psi$-error $\mathrm{er}_D^{\psi,*} = \inf_{\mathbf{f}:\mathcal{X}\rightarrow\mathbb{R}^d} \mathrm{er}_D^\psi[\mathbf{f}]$. A desirable property of $\psi$ is that it be *calibrated* w.r.t. $\ell$, in which case consistency w.r.t. $\psi$ also guarantees consistency w.r.t. $\ell$; we give a formal definition of calibration and statement of this result below.

In what follows, we will denote by $\Delta_n$ the probability simplex in $\mathbb{R}^n$: $\Delta_n = \{\mathbf{p} \in \mathbb{R}_+^n : \sum_i p_i = 1\}$. For $z \in \mathbb{R}$, let $(z)_+ = \max(z, 0)$. We will find it convenient to view the loss function $\ell : \mathcal{Y} \times \mathcal{T} \rightarrow \mathbb{R}_+$ as an $n \times k$ matrix with elements $\ell_{yt} = \ell(y, t)$ for $y \in [n], t \in [k]$, and column vectors $\boldsymbol{\ell}_t = (\ell_{1t}, \ldots, \ell_{nt})^\top \in \mathbb{R}_+^n$ for $t \in [k]$. We will also represent the surrogate loss $\psi : \mathcal{Y} \times \mathbb{R}^d \rightarrow \bar{\mathbb{R}}_+$ as a vector function $\boldsymbol{\psi} : \mathbb{R}^d \rightarrow \bar{\mathbb{R}}_+^n$ with $\psi_y(\mathbf{u}) = \psi(y, \mathbf{u})$ for $y \in [n], \mathbf{u} \in \mathbb{R}^d$, and $\boldsymbol{\psi}(\mathbf{u}) = (\psi_1(\mathbf{u}), \ldots, \psi_n(\mathbf{u}))^\top \in \bar{\mathbb{R}}_+^n$ for $\mathbf{u} \in \mathbb{R}^d$.

**Definition 1** (Calibration). *Let $\ell : \mathcal{Y} \times \mathcal{T} \rightarrow \mathbb{R}_+$ and let $\mathcal{P} \subseteq \Delta_n$. A surrogate loss $\psi : \mathcal{Y} \times \mathbb{R}^d \rightarrow \bar{\mathbb{R}}_+$ is said to be* calibrated *w.r.t. $\ell$ over $\mathcal{P}$ if there exists a function* $\mathrm{pred} : \mathbb{R}^d \rightarrow \mathcal{T}$ *such that*

$$\forall \mathbf{p} \in \mathcal{P} : \quad \inf_{\mathbf{u} \in \mathbb{R}^d : \mathrm{pred}(\mathbf{u}) \notin \mathrm{argmin}_t \mathbf{p}^\top \boldsymbol{\ell}_t} \mathbf{p}^\top \boldsymbol{\psi}(\mathbf{u}) \; > \; \inf_{\mathbf{u} \in \mathbb{R}^d} \mathbf{p}^\top \boldsymbol{\psi}(\mathbf{u}) \, .$$

*In this case we also say $(\psi, \text{pred})$ is $(\ell, \mathcal{P})$-calibrated, or if $\mathcal{P} = \Delta_n$, simply $\ell$-calibrated.*

**Theorem 2** ( [6, 7, 16]). *Let $\ell : \mathcal{Y} \times \mathcal{T} \to \mathbb{R}_+$ and $\psi : \mathcal{Y} \times \mathbb{R}^d \to \bar{\mathbb{R}}_+$. Then $\psi$ is calibrated w.r.t. $\ell$ over $\Delta_n$ iff $\exists$ a function $\text{pred} : \mathbb{R}^d \to \mathcal{T}$ such that for all distributions $D$ on $\mathcal{X} \times \mathcal{Y}$ and all sequences of random (vector) functions $\mathbf{f}_m : \mathcal{X} \to \mathbb{R}^d$ (depending on $(X_1, Y_1), \ldots, (X_m, Y_m)$),*

$$\text{er}_D^\psi[\mathbf{f}_m] \xrightarrow{P} \text{er}_D^{\psi,*} \quad implies \quad \text{er}_D^\ell[\text{pred} \circ \mathbf{f}_m] \xrightarrow{P} \text{er}_D^{\ell,*}.$$

For any instance $x \in \mathcal{X}$, let $\mathbf{p}(x) \in \Delta_n$ denote the conditional label probability vector at $x$, given by $\mathbf{p}(x) = (p_1(x), \ldots, p_n(x))^\top$ where $p_y(x) = \mathbf{P}(Y = y \mid X = x)$. Then one can extend the above result to show that for $\mathcal{P} \subset \Delta_n$, $\psi$ is calibrated w.r.t. $\ell$ over $\mathcal{P}$ iff $\exists$ a function $\text{pred} : \mathbb{R}^d \to \mathcal{T}$ such that the above implication holds for all distributions $D$ on $\mathcal{X} \times \mathcal{Y}$ for which $\mathbf{p}(x) \in \mathcal{P} \ \forall x \in \mathcal{X}$.

**Subset ranking.** Subset ranking problems arise frequently in information retrieval applications. In a subset ranking problem, each instance in $\mathcal{X}$ consists of a query together with a set of say $r$ documents to be ranked. The label space $\mathcal{Y}$ varies from problem to problem: in some cases, labels consist of binary or multi-level relevance judgements for the $r$ documents, in which case $\mathcal{Y} = \{0,1\}^r$ or $\mathcal{Y} = \{0, 1, \ldots, s\}^r$ for some appropriate $s \in \mathbb{Z}_+$; in other cases, labels consist of pairwise preference graphs over the $r$ documents, represented as (possibly weighted) directed acyclic graphs (DAGs) over $r$ nodes. Given examples of such instance-label pairs, the goal is to learn a model to rank documents for new queries/instances; in most cases, the desired ranking takes the form of a permutation over the $r$ documents, so that $\mathcal{T} = S_r$ (where $S_r$ denotes the group of permutations on $r$ objects). As noted earlier, various loss functions are used in practice, and there has been much interest in understanding questions of consistency and calibration for these losses in recent years [9–15, 17]. The focus so far has mostly been on designing $r$-dimensional surrogates, which operate on a surrogate target space of dimension $d = r$; these are also termed 'score-based' surrogates since the resulting algorithms can be viewed as learning one real-valued score function for each of the $r$ documents, and in this case the pred mapping usually consists of simply sorting the documents according to these scores. Below we will apply our result on calibrated surrogates for low-rank loss matrices to obtain new calibrated surrogates – both $r$-dimensional, score-based surrogates and, in some cases, higher-dimensional surrogates – for several subset ranking losses.

## 3 Calibrated Surrogates for Low Rank Loss Matrices

The following is the primary result of our paper. The result gives an explicit construction for a convex, calibrated, least-squares type surrogate loss defined on a low-dimensional surrogate space for any target loss matrix that has a low-rank structure.

**Theorem 3.** *Let $\ell : \mathcal{Y} \times \mathcal{T} \to \mathbb{R}_+$ be a loss function such that there exist $d \in \mathbb{Z}_+$, vectors $\boldsymbol{\alpha}_1, \ldots, \boldsymbol{\alpha}_n \in \mathbb{R}^d$, $\boldsymbol{\beta}_1, \ldots, \boldsymbol{\beta}_k \in \mathbb{R}^d$ and $c \in \mathbb{R}$ such that*

$$\ell(y, t) = \sum_{i=1}^d \alpha_{yi} \beta_{ti} + c.$$

*Let $\psi_\ell^* : \mathcal{Y} \times \mathbb{R}^d \to \bar{\mathbb{R}}_+$ be defined as*

$$\psi_\ell^*(y, \mathbf{u}) = \sum_{i=1}^d (u_i - \alpha_{yi})^2$$

*and let $\text{pred}_\ell^* : \mathbb{R}^d \to \mathcal{T}$ be defined as*

$$\text{pred}_\ell^*(\mathbf{u}) \in \text{argmin}_{t \in [k]} \mathbf{u}^\top \boldsymbol{\beta}_t.$$

*Then $(\psi_\ell^*, \text{pred}_\ell^*)$ is $\ell$-calibrated.*

*Proof.* Let $\mathbf{p} \in \Delta_n$. Define $\mathbf{u}^{\mathbf{P}} \in \mathbb{R}^d$ as $u_i^{\mathbf{P}} = \sum_{y=1}^n p_y \alpha_{yi} \ \forall i \in [d]$. Now for any $\mathbf{u} \in \mathbb{R}^d$, we have

$$\mathbf{p}^\top \boldsymbol{\psi}_\ell^*(\mathbf{u}) = \sum_{i=1}^d \sum_{y=1}^n p_y (u_i - \alpha_{yi})^2.$$

Minimizing this over $\mathbf{u} \in \mathbb{R}^d$ yields that $\mathbf{u}^{\mathbf{P}}$ is the unique minimizer of $\mathbf{p}^\top \boldsymbol{\psi}_\ell^*(\mathbf{u})$. Also, for any $t \in [k]$, we have

$$\mathbf{p}^\top \boldsymbol{\ell}_t \;=\; \sum_{y=1}^{n} p_y \Big( \sum_{i=1}^{d} \alpha_{yi}\beta_{ti} + c \Big) \;=\; (\mathbf{u}^{\mathbf{P}})^\top \boldsymbol{\beta}_t + c\,.$$

Now, for each $t \in [k]$, define
$$\mathrm{regret}^{\ell}_{\mathbf{p}}(t) \;\triangleq\; \mathbf{p}^\top \boldsymbol{\ell}_t - \min_{t'\in[k]} \mathbf{p}^\top \boldsymbol{\ell}_{t'} \;=\; (\mathbf{u}^{\mathbf{P}})^\top \boldsymbol{\beta}_t - \min_{t'\in[k]} (\mathbf{u}^{\mathbf{P}})^\top \boldsymbol{\beta}_{t'}\,.$$

Clearly, by definition of $\mathrm{pred}^*_\ell$, we have $\mathrm{regret}^{\ell}_{\mathbf{p}}(\mathrm{pred}^*_\ell(\mathbf{u}^{\mathbf{P}})) = 0$. Also, if $\mathrm{regret}^{\ell}_{\mathbf{p}}(t) = 0$ for all $t \in [k]$, then trivially $\mathrm{pred}^*_\ell(\mathbf{u}) \in \mathrm{argmin}_t \mathbf{p}^\top \boldsymbol{\ell}_t \; \forall \mathbf{u} \in \mathbb{R}^d$ (and there is nothing to prove in this case). Therefore assume $\exists t \in [k] : \mathrm{regret}^{\ell}_{\mathbf{p}}(t) > 0$, and let
$$\epsilon \;=\; \min_{t\in[k]:\mathrm{regret}^{\ell}_{\mathbf{p}}(t)>0} \mathrm{regret}^{\ell}_{\mathbf{p}}(t)\,.$$

Then we have
$$\inf_{\mathbf{u}\in\mathbb{R}^d:\mathrm{pred}^*_\ell(\mathbf{u})\notin\mathrm{argmin}_t \mathbf{p}^\top\boldsymbol{\ell}_t} \mathbf{p}^\top \boldsymbol{\psi}^*_\ell(\mathbf{u}) \;=\; \inf_{\mathbf{u}\in\mathbb{R}^d:\mathrm{regret}^{\ell}_{\mathbf{p}}(\mathrm{pred}^*_\ell(\mathbf{u}))\geq\epsilon} \mathbf{p}^\top \boldsymbol{\psi}^*_\ell(\mathbf{u})$$
$$=\; \inf_{\mathbf{u}\in\mathbb{R}^d:\mathrm{regret}^{\ell}_{\mathbf{p}}(\mathrm{pred}^*_\ell(\mathbf{u}))\geq\mathrm{regret}^{\ell}_{\mathbf{p}}(\mathrm{pred}^*_\ell(\mathbf{u}^{\mathbf{P}}))+\epsilon} \mathbf{p}^\top \boldsymbol{\psi}^*_\ell(\mathbf{u})\,.$$

Now, we claim that the mapping $\mathbf{u} \mapsto \mathrm{regret}^{\ell}_{\mathbf{p}}(\mathrm{pred}^*_\ell(\mathbf{u}))$ is continuous at $\mathbf{u} = \mathbf{u}^{\mathbf{P}}$. To see this, suppose the sequence $\{\mathbf{u}_m\}$ converges to $\mathbf{u}^{\mathbf{P}}$. Then we have
$$\mathrm{regret}^{\ell}_{\mathbf{p}}(\mathrm{pred}^*_\ell(\mathbf{u}_m)) \;=\; (\mathbf{u}^{\mathbf{P}})^\top \boldsymbol{\beta}_{\mathrm{pred}^*_\ell(\mathbf{u}_m)} - \min_{t'\in[k]} (\mathbf{u}^{\mathbf{P}})^\top \boldsymbol{\beta}_{t'}$$
$$=\; (\mathbf{u}^{\mathbf{P}} - \mathbf{u}_m)^\top \boldsymbol{\beta}_{\mathrm{pred}^*_\ell(\mathbf{u}_m)} + \mathbf{u}_m^\top \boldsymbol{\beta}_{\mathrm{pred}^*_\ell(\mathbf{u}_m)} - \min_{t'\in[k]} (\mathbf{u}^{\mathbf{P}})^\top \boldsymbol{\beta}_{t'}$$
$$=\; (\mathbf{u}^{\mathbf{P}} - \mathbf{u}_m)^\top \boldsymbol{\beta}_{\mathrm{pred}^*_\ell(\mathbf{u}_m)} + \min_{t'\in[k]} \mathbf{u}_m^\top \boldsymbol{\beta}_{t'} - \min_{t'\in[k]} (\mathbf{u}^{\mathbf{P}})^\top \boldsymbol{\beta}_{t'}$$

The last equality holds by definition of $\mathrm{pred}^*_\ell$. It is easy to see the term on the right goes to zero as $\mathbf{u}_m$ converges to $\mathbf{u}^{\mathbf{P}}$. Thus $\mathrm{regret}^{\ell}_{\mathbf{p}}(\mathrm{pred}^*_\ell(\mathbf{u}_m))$ converges to $\mathrm{regret}^{\ell}_{\mathbf{p}}(\mathrm{pred}^*_\ell(\mathbf{u}^{\mathbf{P}})) = 0$, yielding continuity at $\mathbf{u}^{\mathbf{P}}$. In particular, this implies $\exists \delta > 0$ such that
$$\|\mathbf{u} - \mathbf{u}^{\mathbf{P}}\| < \delta \implies \mathrm{regret}^{\ell}_{\mathbf{p}}(\mathrm{pred}^*_\ell(\mathbf{u})) - \mathrm{regret}^{\ell}_{\mathbf{p}}(\mathrm{pred}^*_\ell(\mathbf{u}^{\mathbf{P}})) < \epsilon\,.$$

This gives
$$\inf_{\mathbf{u}\in\mathbb{R}^d:\mathrm{regret}^{\ell}_{\mathbf{p}}(\mathrm{pred}^*_\ell(\mathbf{u}))\geq\mathrm{regret}^{\ell}_{\mathbf{p}}(\mathrm{pred}^*_\ell(\mathbf{u}^{\mathbf{P}}))+\epsilon} \mathbf{p}^\top \boldsymbol{\psi}^*_\ell(\mathbf{u}) \;\geq\; \inf_{\mathbf{u}\in\mathbb{R}^d:\|\mathbf{u}-\mathbf{u}^{\mathbf{P}}\|\geq\delta} \mathbf{p}^\top \boldsymbol{\psi}^*_\ell(\mathbf{u})$$
$$>\; \inf_{\mathbf{u}\in\mathbb{R}^d} \mathbf{p}^\top \boldsymbol{\psi}^*_\ell(\mathbf{u})\,,$$

where the last inequality holds since $\mathbf{p}^\top \boldsymbol{\psi}^*_\ell(\mathbf{u})$ is a strictly convex function of $\mathbf{u}$ and $\mathbf{u}^{\mathbf{P}}$ is its unique minimizer. The above sequence of inequalities give us that
$$\inf_{\mathbf{u}\in\mathbb{R}^d:\mathrm{pred}^*_\ell(\mathbf{u})\notin\mathrm{argmin}_t \mathbf{p}^\top\boldsymbol{\ell}_t} \mathbf{p}^\top \boldsymbol{\psi}^*_\ell(\mathbf{u}) \;>\; \inf_{\mathbf{u}\in\mathbb{R}^d} \mathbf{p}^\top \boldsymbol{\psi}^*_\ell(\mathbf{u})\,.$$

Since this holds for all $\mathbf{p} \in \Delta_n$, we have that $(\boldsymbol{\psi}^*_\ell, \mathrm{pred}^*_\ell)$ is $\ell$-calibrated. $\qquad\square$

We note that Ramaswamy and Agarwal [16] showed a similar least-squares type surrogate calibrated for any loss $\ell : \mathcal{Y} \times \mathcal{T} \rightarrow \mathbb{R}_+$; indeed our proof technique above draws inspiration from the proof technique there. However, the surrogate they gave was defined on a surrogate space of dimension $n-1$, where $n$ is the number of class labels in $\mathcal{Y}$. For many practical problems, this is an intractably large number. For example, as noted above, in the subset ranking problems we consider, the number of class labels is typically exponential in $r$, the number of documents associated with each query. On the other hand, as we will see below, many subset ranking losses have a low-rank structure, with rank linear or quadratic in $r$, allowing us to use the above result to design convex calibrated surrogates on an $O(r)$ or $O(r^2)$-dimensional space. Ramaswamy and Agarwal also gave another result in which they showed that any loss matrix of rank $d$ has a $d$-dimensional convex calibrated surrogate; however the surrogate there was defined such that it took values $< \infty$ on an awkward space in $\mathbb{R}^d$ (not the full space $\mathbb{R}^d$) that would be difficult to construct in practice, and moreover, their result did not yield an explicit construction for the pred mapping required to use a calibrated surrogate in practice. Our result above combines the benefits of both these previous results, allowing explicit construction of low-dimensional least-squares type surrogates for any low-rank loss matrix. The following sections will illustrate several applications of this result.

# 4 Calibrated Surrogates for Precision@$q$

The Precision@$q$ is a popular performance measure for subset ranking problems in information retrieval. As noted above, in a subset ranking problem, each instance in $\mathcal{X}$ consists of a query together with a set of $r$ documents to be ranked. Consider a setting with binary relevance judgement labels, so that $\mathcal{Y} = \{0, 1\}^r$ with $n = 2^r$. The prediction space is $\mathcal{T} = S_r$ (group of permutations on $r$ objects) with $k = r!$. For $\mathbf{y} \in \{0, 1\}^r$ and $\sigma \in S_r$, where $\sigma(i)$ denotes the position of document $i$ under $\sigma$, the Precision@$q$ loss for any integer $q \in [r]$ can be written as follows:

$$
\begin{aligned}
\ell_{\mathrm{P@}q}(\mathbf{y}, \sigma) &= 1 - \frac{1}{q} \sum_{i=1}^{q} y_{\sigma^{-1}(i)} \\
&= 1 - \frac{1}{q} \sum_{i=1}^{r} y_i \cdot \mathbf{1}(\sigma(i) \leq q) \,.
\end{aligned}
$$

Therefore, by Theorem 3, for the $r$-dimensional surrogate $\psi_{\mathrm{P@}q}^* : \{0, 1\}^r \times \mathbb{R}^r \to \bar{\mathbb{R}}_+$ and $\mathrm{pred}_{\mathrm{P@}q}^* : \mathbb{R}^r \to S_r$ defined as

$$
\psi_{\mathrm{P@}q}^*(\mathbf{y}, \mathbf{u}) = \sum_{i=1}^{r} (u_i - y_i)^2
$$

$$
\mathrm{pred}_{\mathrm{P@}q}^*(\mathbf{u}) \in \mathrm{argmax}_{\sigma \in S_r} \sum_{i=1}^{r} u_i \cdot \mathbf{1}(\sigma(i) \leq q) \,,
$$

we have that $(\psi_{\mathrm{P@}q}^*, \mathrm{pred}_{\mathrm{P@}q}^*)$ is $\ell_{\mathrm{P@}q}$-calibrated. It can easily be seen that for any $\mathbf{u} \in \mathbb{R}^r$, any permutation $\sigma$ which places the top $q$ documents sorted in decreasing order of scores $u_i$ in the top $q$ positions achieves the maximum in $\mathrm{pred}_{\mathrm{P@}q}^*(\mathbf{u})$; thus $\mathrm{pred}_{\mathrm{P@}q}^*(\mathbf{u})$ can be implemented efficiently using a standard sorting or selection algorithm. Note that the popular winner-take-all (WTA) loss, which assigns a loss of 0 if the top-ranked item is relevant (i.e. if $y_{\sigma^{-1}(1)} = 1$) and 1 otherwise, is simply a special case of the above loss with $q = 1$; therefore the above construction also yields a calibrated surrogate for the WTA loss. To our knowledge, this is the first example of convex, calibrated surrogates for the Precision@$q$ and WTA losses.

# 5 Calibrated Surrogates for Expected Rank Utility

The expected rank utility (ERU) is a popular subset ranking performance measure used in recommender systems displaying short ranked lists [18]. In this case the labels consist of multi-level relevance judgements (such as 0 to 5 stars), so that $\mathcal{Y} = \{0, 1, \ldots, s\}^r$ for some appropriate $s \in \mathbb{Z}_+$ with $n = (s + 1)^r$. The prediction space again is $\mathcal{T} = S_r$ with $k = r!$. For $\mathbf{y} \in \{0, 1, \ldots, s\}^r$ and $\sigma \in S_r$, where $\sigma(i)$ denotes the position of document $i$ under $\sigma$, the ERU loss is defined as

$$
\ell_{\mathrm{ERU}}(\mathbf{y}, \sigma) = z - \sum_{i=1}^{r} \max(y_i - v, 0) \cdot 2^{\frac{1 - \sigma(i)}{w - 1}} \,,
$$

where $z$ is a constant to ensure the positivity of the loss, $v \in [s]$ is a constant that indicates a neutral score, and $w \in \mathbb{R}$ is a constant indicating the viewing half-life. Thus, by Theorem 3, for the $r$-dimensional surrogate $\psi_{\mathrm{ERU}}^* : \{0, 1, \ldots, s\}^r \times \mathbb{R}^r \to \bar{\mathbb{R}}_+$ and $\mathrm{pred}_{\mathrm{ERU}}^* : \mathbb{R}^r \to S_r$ defined as

$$
\psi_{\mathrm{ERU}}^*(\mathbf{y}, \mathbf{u}) = \sum_{i=1}^{r} (u_i - \max(y_i - v, 0))^2
$$

$$
\mathrm{pred}_{\mathrm{ERU}}^*(\mathbf{u}) \in \mathrm{argmax}_{\sigma \in S_r} \sum_{i=1}^{r} u_i \cdot 2^{\frac{1 - \sigma(i)}{w - 1}} \,,
$$

we have that $(\psi_{\mathrm{ERU}}^*, \mathrm{pred}_{\mathrm{ERU}}^*)$ is $\ell_{\mathrm{ERU}}$-calibrated. It can easily be seen that for any $\mathbf{u} \in \mathbb{R}^r$, any permutation $\sigma$ satisfying the condition

$$
u_i > u_j \implies \sigma(i) < \sigma(j)
$$

achieves the maximum in $\mathrm{pred}_{\mathrm{ERU}}^*(\mathbf{u})$, and therefore $\mathrm{pred}_{\mathrm{ERU}}^*(\mathbf{u})$ can be implemented efficiently by simply sorting the $r$ documents in decreasing order of scores $u_i$. As for Precision@$q$, to our knowledge, this is the first example of a convex, calibrated surrogate for the ERU loss.

# 6  Calibrated Surrogates for Mean Average Precision

The mean average precision (MAP) is a widely used ranking performance measure in information retrieval and related applications [15, 19]. As with the Precision@q loss, $\mathcal{Y} = \{0,1\}^r$ and $\mathcal{T} = S_r$. For $\mathbf{y} \in \{0,1\}^r$ and $\sigma \in S_r$, where $\sigma(i)$ denotes the position of document $i$ under $\sigma$, the MAP loss is defined as follows:

$$\ell_{\text{MAP}}(\mathbf{y}, \sigma) = 1 - \frac{1}{|\{\gamma : y_\gamma = 1\}|} \sum_{i:y_i=1} \frac{1}{\sigma(i)} \sum_{j=1}^{\sigma(i)} y_{\sigma^{-1}(j)} \,.$$

It was recently shown that there cannot exist any $r$-dimensional convex, calibrated surrogates for the MAP loss [15]. We now re-write the MAP loss above in a manner that allows us to show the existence of an $O(r^2)$-dimensional convex, calibrated surrogate. In particular, we can write

$$\ell_{\text{MAP}}(\mathbf{y}, \sigma) = 1 - \frac{1}{\sum_{\gamma=1}^r y_\gamma} \sum_{i=1}^r \sum_{j=1}^i \frac{y_{\sigma^{-1}(i)} y_{\sigma^{-1}(j)}}{i} \, . \; = \; 1 - \frac{1}{\sum_{\gamma=1}^r y_\gamma} \sum_{i=1}^r \sum_{j=1}^i \frac{y_i y_j}{\max(\sigma(i), \sigma(j))}$$

Thus, by Theorem 3, for the $\frac{r(r+1)}{2}$-dimensional surrogate $\psi_{\text{MAP}}^* : \{0,1\}^r \times \mathbb{R}^{r(r+1)/2} \to \bar{\mathbb{R}}_+$ and $\text{pred}_{\text{MAP}}^* : \mathbb{R}^{r(r+1)/2} \to S_r$ defined as

$$\psi_{\text{MAP}}^*(\mathbf{y}, \mathbf{u}) = \sum_{i=1}^r \sum_{j=1}^i \left( u_{ij} - \frac{y_i y_j}{\sum_{\gamma=1}^r y_\gamma} \right)^2$$

$$\text{pred}_{\text{MAP}}^*(\mathbf{u}) \in \text{argmax}_{\sigma \in S_r} \sum_{i=1}^r \sum_{j=1}^i u_{ij} \cdot \frac{1}{\max(\sigma(i), \sigma(j))} \,,$$

we have that $(\psi_{\text{MAP}}^*, \text{pred}_{\text{MAP}}^*)$ is $\ell_{\text{MAP}}$-calibrated.

Note however that the optimization problem associated with computing $\text{pred}_{\text{MAP}}^*(\mathbf{u})$ above can be written as a quadratic assignment problem (QAP), and most QAPs are known to be NP-hard. We conjecture that the QAP associated with the mapping $\text{pred}_{\text{MAP}}^*$ above is also NP-hard. Therefore, while the surrogate loss $\psi_{\text{MAP}}^*$ is calibrated for $\ell_{\text{MAP}}$ and can be minimized efficiently over a training sample to learn a model $\mathbf{f} : \mathcal{X} \to \mathbb{R}^{r(r+1)/2}$, for large $r$, evaluating the mapping required to transform predictions in $\mathbb{R}^{r(r+1)/2}$ back to predictions in $S_r$ is likely to be computationally infeasible. Below we describe an alternate mapping in place of $\text{pred}_{\text{MAP}}^*$ which can be computed efficiently, and show that under certain conditions on the probability distribution, the surrogate $\psi_{\text{MAP}}^*$ together with this mapping is still calibrated for $\ell_{\text{MAP}}$.

Specifically, define $\overline{\text{pred}}_{\text{MAP}} : \mathbb{R}^{r(r+1)/2} \to S_r$ as follows:

$$\overline{\text{pred}}_{\text{MAP}}(\mathbf{u}) \in \left\{ \sigma \in S_r : u_{ii} > u_{jj} \implies \sigma(i) < \sigma(j) \right\}.$$

Clearly, $\overline{\text{pred}}_{\text{MAP}}(\mathbf{u})$ can be implemented efficiently by simply sorting the 'diagonal' elements $u_{ii}$ for $i \in [r]$. Also, let $\Delta_{\mathcal{Y}}$ denote the probability simplex over $\mathcal{Y}$, and for each $\mathbf{p} \in \Delta_{\mathcal{Y}}$, define $\mathbf{u}^{\mathbf{P}} \in \mathbb{R}^{r(r+1)/2}$ as follows:

$$u_{ij}^{\mathbf{P}} = \mathbf{E}_{Y \sim \mathbf{p}} \left[ \frac{Y_i Y_j}{\sum_{\gamma=1}^r Y_\gamma} \right] = \sum_{\mathbf{y} \in \mathcal{Y}} p_{\mathbf{y}} \left( \frac{y_i y_j}{\sum_{\gamma=1}^r y_\gamma} \right) \quad \forall i, j \in [r] : i \geq j \,.$$

Now define $\mathcal{P}_{\text{reinforce}} \subset \Delta_{\mathcal{Y}}$ as follows:

$$\mathcal{P}_{\text{reinforce}} = \left\{ \mathbf{p} \in \Delta_{\mathcal{Y}} : u_{ii}^{\mathbf{P}} \geq u_{jj}^{\mathbf{P}} \implies u_{ii}^{\mathbf{P}} \geq u_{jj}^{\mathbf{P}} + \sum_{\gamma \in [r] \setminus \{i,j\}} (u_{j\gamma}^{\mathbf{P}} - u_{i\gamma}^{\mathbf{P}})_+ \right\},$$

where we set $u_{ij}^{\mathbf{P}} = u_{ji}^{\mathbf{P}}$ for $i < j$. Then we have the following result:

**Theorem 4.** $(\psi_{\text{MAP}}^*, \overline{\text{pred}}_{\text{MAP}})$ is $(\ell_{\text{MAP}}, \mathcal{P}_{\text{reinforce}})$-calibrated.

The ideal predictor $\text{pred}_{\text{MAP}}^*$ uses the entire $\mathbf{u}$ matrix, but the predictor $\overline{\text{pred}}_{\text{MAP}}$, uses only the diagonal elements. The noise conditions $\mathcal{P}_{\text{reinforce}}$ can be viewed as basically enforcing that the diagonal elements dominate and enforce a clear ordering themselves.

In fact, since the mapping $\overline{\text{pred}}_{\text{MAP}}$ depends on only the diagonal elements of $\mathbf{u}$, we can equivalently define an $r$-dimensional surrogate that is calibrated w.r.t. $\ell_{\text{MAP}}$ over $\mathcal{P}_{\text{reinforce}}$. Specifically, we have the following immediate corollary:

**Corollary 5.** *Let* $\widetilde{\psi}_{\mathrm{MAP}} : \{0,1\}^r \times \mathbb{R}^r \rightarrow \bar{\mathbb{R}}_+$ *and* $\widetilde{\mathrm{pred}}_{\mathrm{MAP}} : \mathbb{R}^r \rightarrow S_r$ *be defined as*

$$\widetilde{\psi}_{\mathrm{MAP}}(\mathbf{y}, \widetilde{\mathbf{u}}) = \sum_{i=1}^{r} \left( \widetilde{u}_i - \frac{y_i}{\sum_{\gamma=1}^{r} y_\gamma} \right)^2$$

$$\widetilde{\mathrm{pred}}_{\mathrm{MAP}}(\widetilde{\mathbf{u}}) \in \left\{ \sigma \in S_r : \widetilde{u}_i > \widetilde{u}_j \implies \sigma(i) < \sigma(j) \right\}.$$

*Then* $(\widetilde{\psi}_{\mathrm{MAP}}, \widetilde{\mathrm{pred}}_{\mathrm{MAP}})$ *is* $(\ell_{\mathrm{MAP}}, \mathcal{P}_{\mathrm{reinforce}})$-*calibrated.*

Looking at the form of $\widetilde{\psi}_{\mathrm{MAP}}$ and $\widetilde{\mathrm{pred}}_{\mathrm{MAP}}$, we can see that the function $\mathbf{s} : \mathcal{Y} \rightarrow \mathbb{R}^r$ defined as $s_i(\mathbf{y}) = y_i / (\sum_{\gamma=1}^{r} y_r)$ is a 'standardization function' for the MAP loss over $\mathcal{P}_{\mathrm{reinforce}}$, and therefore it follows that any 'order-preserving surrogate' with this standardization function is also calibrated with the MAP loss over $\mathcal{P}_{\mathrm{reinforce}}$ [13]. To our knowledge, this is the first example of conditions on the probability distribution under which a convex calibrated (and moreover, score-based) surrogate can be designed for the MAP loss.

## 7 Calibrated Surrogates for Pairwise Disagreement

The pairwise disagreement (PD) loss is a natural and widely used loss in subset ranking [11, 17]. The label space $\mathcal{Y}$ here consists of a finite number of (possibly weighted) directed acyclic graphs (DAGs) over $r$ nodes; we can represent each such label as a vector $\mathbf{y} \in \mathbb{R}_+^{r(r-1)}$ where at least one of $y_{ij}$ or $y_{ji}$ is 0 for each $i \neq j$, with $y_{ij} > 0$ indicating a preference for document $i$ over document $j$ and $y_{ij}$ denoting the weight of the preference. The prediction space as usual is $\mathcal{T} = S_r$ with $k = r!$. For $\mathbf{y} \in \mathcal{Y}$ and $\sigma \in S_r$, where $\sigma(i)$ denotes the position of document $i$ under $\sigma$, the PD loss is defined as follows:

$$\ell_{\mathrm{PD}}(\mathbf{y}, \sigma) = \sum_{i=1}^{r} \sum_{j \neq i} y_{ij} \, \mathbf{1}\big(\sigma(i) > \sigma(j)\big).$$

It was recently shown that there cannot exist any $r$-dimensional convex, calibrated surrogates for the PD loss [15, 16]. By Theorem 3, for the $r(r-1)$-dimensional surrogate $\psi_{\mathrm{PD}}^* : \mathcal{Y} \times \mathbb{R}^{r(r-1)} \rightarrow \bar{\mathbb{R}}_+$ and $\mathrm{pred}_{\mathrm{PD}}^* : \mathbb{R}^{r(r-1)} \rightarrow S_r$ defined as

$$\psi_{\mathrm{PD}}^*(\mathbf{y}, \mathbf{u}) = \sum_{i=1}^{r} \sum_{j \neq i} (u_{ij} - y_{ij})^2 \tag{1}$$

$$\mathrm{pred}_{\mathrm{PD}}^*(\mathbf{u}) \in \mathrm{argmin}_{\sigma \in S_r} \sum_{i=1}^{r} \sum_{j \neq i} u_{ij} \cdot \mathbf{1}\big(\sigma(i) > \sigma(j)\big)$$

we immediately have that $(\psi_{\mathrm{PD}}^*, \mathrm{pred}_{\mathrm{PD}}^*)$ is $\ell_{\mathrm{PD}}$-calibrated (in fact the loss matrix $\ell_{\mathrm{PD}}$ has rank at most $\frac{r(r-1)}{2}$, allowing for an $\frac{r(r-1)}{2}$-dimensional surrogate; we use $r(r-1)$ dimensions for convenience).

In this case, the optimization problem associated with computing $\mathrm{pred}_{\mathrm{PD}}^*(\mathbf{u})$ above is a minimum weighted feedback arc set (MWFAS) problem, which is known to be NP-hard. Therefore, as with the MAP loss, while the surrogate loss $\psi_{\mathrm{PD}}^*$ is calibrated for $\ell_{\mathrm{PD}}$ and can be minimized efficiently over a training sample to learn a model $\mathbf{f} : \mathcal{X} \rightarrow \mathbb{R}^{r(r-1)}$, for large $r$, evaluating the mapping required to transform predictions in $\mathbb{R}^{r(r-1)}$ back to predictions in $S_r$ is likely to be computationally infeasible.

Below we give two sets of results. In Section 7.1, we give a family of score-based ($r$-dimensional) surrogates that are calibrated with the PD loss under different conditions on the probability distribution; these surrogates and conditions generalize those of Duchi et al. [11]. In Section 7.2, we give a different condition on the probability distribution under which we can actually avoid 'difficult' graphs being passed to $\mathrm{pred}_{\mathrm{PD}}^*$. This condition is more general (i.e. encompasses a larger set of probability distributions) than those associated with the score-based surrogates; this gives a new (non-score-based, $r(r-1)$-dimensional) surrogate with an efficiently computable pred mapping that is calibrated with the PD loss over a larger set of probability distributions than previous surrogates for this loss.

### 7.1 Family of $r$-Dimensional Surrogates Calibrated with $\ell_{\mathrm{PD}}$ Under Noise Conditions

The following gives a family of score-based surrogates, parameterized by functions $\mathbf{f} : \mathcal{Y} \rightarrow \mathbb{R}^r$, that are calibrated with the PD loss under different conditions on the probability distribution:

**Theorem 6.** *Let* $\mathbf{f} : \mathcal{Y} \rightarrow \mathbb{R}^r$ *be any function that maps DAGs* $\mathbf{y} \in \mathcal{Y}$ *to score vectors* $\mathbf{f}(\mathbf{y}) \in \mathbb{R}^r$. *Let* $\psi_\mathbf{f} : \mathcal{Y} \times \mathbb{R}^r \rightarrow \bar{\mathbb{R}}_+$, $\mathrm{pred} : \mathbb{R}^r \rightarrow S_r$ *and* $\mathcal{P}_\mathbf{f} \subset \Delta_\mathcal{Y}$ *be defined as*

$$\psi_\mathbf{f}(\mathbf{y}, \mathbf{u}) = \sum_{i=1}^{r} \big(u_i - f_i(\mathbf{y})\big)^2$$

$$\mathrm{pred}(\mathbf{u}) \in \Big\{ \sigma \in S_r : u_i > u_j \implies \sigma(i) < \sigma(j) \Big\}$$

$$\mathcal{P}_\mathbf{f} = \Big\{ \mathbf{p} \in \Delta_\mathcal{Y} : \mathbf{E}_{Y \sim \mathbf{p}}[Y_{ij}] > \mathbf{E}_{Y \sim \mathbf{p}}[Y_{ji}] \implies \mathbf{E}_{Y \sim \mathbf{p}}[f_i(Y)] > \mathbf{E}_{Y \sim \mathbf{p}}[f_j(Y)] \Big\}.$$

*Then* $(\psi_\mathbf{f}, \mathrm{pred})$ *is* $(\ell_{\mathrm{PD}}, \mathcal{P}_\mathbf{f})$-*calibrated.*

The noise conditions $\mathcal{P}_\mathbf{f}$ state that the expected value of function $\mathbf{f}$ must decide the 'right' ordering. We note that the surrogate given by Duchi et al. [11] can be written in our notation as

$$\psi_{\mathrm{DMJ}}(\mathbf{y}, \mathbf{u}) = \sum_{i=1}^{r} \sum_{j \neq i} y_{ij}(u_j - u_i) + \nu \sum_{i=1}^{r} \lambda(u_i),$$

where $\lambda$ is a strictly convex and 1-coercive function and $\nu > 0$. Taking $\lambda(z) = z^2$ and $\nu = \frac{1}{2}$ gives a special case of the family of score-based surrogates in Theorem 6 above obtained by taking $\mathbf{f}$ as

$$f_i(\mathbf{y}) = \sum_{j \neq i}(y_{ij} - y_{ji}).$$

Indeed, the set of noise conditions under which the surrogate $\psi_{\mathrm{DMJ}}$ is shown to be calibrated with the PD loss in Duchi et al. [11] is exactly the set $\mathcal{P}_\mathbf{f}$ above with this choice of $\mathbf{f}$. We also note that $\mathbf{f}$ can be viewed as a 'standardization function' [13] for the PD loss over $\mathcal{P}_\mathbf{f}$.

## 7.2 An $O(r^2)$-dimensional Surrogate Calibrated with $\ell_{\mathrm{PD}}$ Under More General Conditions

Consider now the $r(r-1)$-dimensional surrogate $\psi_{\mathrm{PD}}^* : \mathcal{Y} \times \mathbb{R}^{r(r-1)}$ defined in Eq. (1). We noted the corresponding mapping $\mathrm{pred}_{\mathrm{PD}}^*$ involved an NP-hard optimization problem. Here we give an alternate mapping $\overline{\mathrm{pred}}_{\mathrm{PD}} : \mathbb{R}^{r(r-1)} \rightarrow S_r$ that can be computed efficiently, and show that under certain conditions on the probability distribution , the surrogate $\psi_{\mathrm{PD}}^*$ together with this mapping $\overline{\mathrm{pred}}_{\mathrm{PD}}$ is calibrated for $\ell_{\mathrm{PD}}$. The mapping $\overline{\mathrm{pred}}_{\mathrm{PD}}$ is described by Algorithm 1 below:

---

**Algorithm 1** $\overline{\mathrm{pred}}_{\mathrm{PD}}$ **(Input:** $\mathbf{u} \in \mathbb{R}^{r(r-1)}$; **Output:** Permutation $\sigma \in S_r$)

---

Construct a directed graph over $[r]$ with edge $(i,j)$ having weight $(u_{ij} - u_{ji})_+$. If this graph is acyclic, return any topological sorted order. If the graph has cycles, sort the edges in ascending order by weight and delete them one by one (smallest weight first) until the graph becomes acyclic; return any topological sorted order of the resulting acyclic graph.

---

For each $\mathbf{p} \in \Delta_\mathcal{Y}$, define $E^\mathbf{p} = \{(i,j) \in [r] \times [r] : \mathbf{E}_{Y \sim \mathbf{p}}[Y_{ij}] > \mathbf{E}_{Y \sim \mathbf{p}}[Y_{ji}]\}$, and define

$$\mathcal{P}_{\mathrm{DAG}} = \Big\{ \mathbf{p} \in \Delta_\mathcal{Y} : \big([r], E^\mathbf{p}\big) \text{ is a DAG} \Big\}.$$

Then we have the following result:

**Theorem 7.** $(\psi_{\mathrm{PD}}^*, \overline{\mathrm{pred}}_{\mathrm{PD}})$ *is* $(\ell_{\mathrm{PD}}, \mathcal{P}_{\mathrm{DAG}})$-*calibrated.*

It is easy to see that $\mathcal{P}_{\mathrm{DAG}} \supsetneq \mathcal{P}_\mathbf{f}\ \forall \mathbf{f}$ (where $\mathcal{P}_\mathbf{f}$ is as defined in Theorem 6), so that the above result yields a low-dimensional, convex surrogate with an efficiently computable pred mapping that is calibrated for the PD loss under a broader set of conditions than the previous surrogates.

## 8 Conclusion

Calibration of surrogate losses is an important property in designing consistent learning algorithms. We have given an explicit method for constructing calibrated surrogates for any learning problem with a low-rank loss structure, and have used this to obtain several new results for subset ranking, including new calibrated surrogates for the Precision@$q$, ERU, MAP and PD losses.

**Acknowledgments**

The authors thank the anonymous reviewers, Aadirupa Saha and Shiv Ganesh for their comments. HGR acknowledges a Tata Consultancy Services (TCS) PhD fellowship and the Indo-US Virtual Institute for Mathematical and Statistical Sciences (VIMSS). SA thanks the Department of Science & Technology (DST) and Indo-US Science & Technology Forum (IUSSTF) for their support. AT gratefully acknowledges the support of NSF under grant IIS-1319810.

## Footnotes

[1] Here $\xrightarrow{\mathrm{P}}$ denotes convergence in probability: $X_m \xrightarrow{\mathrm{P}} a$ if $\forall \epsilon > 0, \mathbf{P}(|X_m - a| \geq \epsilon) \rightarrow 0$ as $m \rightarrow \infty$.

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
