[Supplementary Material]

# Convex Calibrated Surrogates for Low-Rank Loss Matrices with Applications to Subset Ranking Losses

## Appendix

### Proof of Theorem 4

*Proof.* Let $\mathbf{p} \in \mathcal{P}_{\text{reinforce}}$. We define $\mathbf{u}^{\mathbf{P}} \in \mathbb{R}^{r(r+1)/2}$ again here for convenience:

$$u_{ij}^{\mathbf{P}} \;=\; \mathbf{E}_{Y \sim \mathbf{p}}\left[\frac{Y_i Y_j}{\sum_{\gamma=1}^{r} Y_\gamma}\right] \;=\; \sum_{\mathbf{y} \in \mathcal{Y}} p_{\mathbf{y}}\left(\frac{y_i y_j}{\sum_{\gamma=1}^{r} y_\gamma}\right) \quad \forall i, j \in [r] : i \geq j\,.$$

It is easy to see that $\mathbf{u}^{\mathbf{P}} \in \mathbb{R}^{r(r+1)/2}$ is the unique minimizer of $\mathbf{p}^\top \boldsymbol{\psi}_{\text{MAP}}^*(\mathbf{u})$ over $\mathbf{u} \in \mathbb{R}^{r(r+1)/2}$.

Recall also that while $u_{ij}^{\mathbf{P}}$ above is defined only for $i \geq j$, we also set $u_{ij}^{\mathbf{P}} = u_{ji}^{\mathbf{P}}$ for $i < j$.

For brevity, we will write $\ell_{\text{MAP}}$ as $\ell$ below. We have from the definition of the MAP loss,

$$
\begin{aligned}
\mathbf{p}^\top \boldsymbol{\ell}_\sigma \;&=\; 1 - \sum_{i=1}^{r} \sum_{j=1}^{i} u_{ij}^{\mathbf{P}} \frac{1}{\max(\sigma(i), \sigma(j))} \\
&=\; 1 - \sum_{i=1}^{r} \frac{1}{i} \sum_{j=1}^{i} u_{\sigma^{-1}(i)\sigma^{-1}(j)}^{\mathbf{P}}\,.
\end{aligned}
\tag{2}
$$

Now define the following sets:

$$\Pi^*(\mathbf{p}) = \operatorname{argmin}_{\sigma \in S_r} \mathbf{p}^\top \boldsymbol{\ell}_\sigma$$

$$\Pi(\mathbf{p}) = \left\{\sigma \in S_r : u_{ii}^{\mathbf{P}} > u_{jj}^{\mathbf{P}} \implies \sigma(i) < \sigma(j)\right\}.$$

From Lemma 8 below, we have that $\Pi(\mathbf{p}) \subseteq \Pi^*(\mathbf{p})$.

By the definition of $\overline{\text{pred}}_{\text{MAP}}$ and $\Pi(\mathbf{p})$, we also have that $\exists \epsilon > 0$ such that for any $\mathbf{u} \in \mathbb{R}^{r(r+1)/2}$,

$$\|\mathbf{u} - \mathbf{u}^{\mathbf{P}}\| < \epsilon \implies \overline{\text{pred}}_{\text{MAP}}(\mathbf{u}) \in \Pi(\mathbf{p})\,.$$

Thus, we have

$$
\begin{aligned}
\inf_{\mathbf{u} \in \mathbb{R}^{r(r+1)/2} : \overline{\text{pred}}_{\text{MAP}}(\mathbf{u}) \notin \operatorname{argmin}_\sigma \mathbf{p}^\top \boldsymbol{\ell}_\sigma} \mathbf{p}^\top \boldsymbol{\psi}_{\text{MAP}}^*(\mathbf{u}) \;&=\; \inf_{\mathbf{u} \in \mathbb{R}^{r(r+1)/2} : \overline{\text{pred}}_{\text{MAP}}(\mathbf{u}) \notin \Pi^*(\mathbf{p})} \mathbf{p}^\top \boldsymbol{\psi}_{\text{MAP}}^*(\mathbf{u}) \\
&\geq\; \inf_{\mathbf{u} \in \mathbb{R}^{r(r+1)/2} : \overline{\text{pred}}_{\text{MAP}}(\mathbf{u}) \notin \Pi(\mathbf{p})} \mathbf{p}^\top \boldsymbol{\psi}_{\text{MAP}}^*(\mathbf{u}) \\
&\geq\; \inf_{\mathbf{u} \in \mathbb{R}^{r(r+1)/2} : \|\mathbf{u} - \mathbf{u}^{\mathbf{P}}\| \geq \epsilon} \mathbf{p}^\top \boldsymbol{\psi}_{\text{MAP}}^*(\mathbf{u}) \\
&>\; \inf_{\mathbf{u} \in \mathbb{R}^{r(r+1)/2}} \mathbf{p}^\top \boldsymbol{\psi}_{\text{MAP}}^*(\mathbf{u})\,,
\end{aligned}
$$

where the last inequality follows since $\mathbf{p}^\top \boldsymbol{\psi}_{\text{MAP}}^*(\mathbf{u})$ is a strictly convex function of $\mathbf{u}$ and $\mathbf{u}^{\mathbf{P}}$ is its unique minimizer.

Since the above holds for all $\mathbf{p} \in \mathcal{P}_{\text{reinforce}}$, we have that $(\boldsymbol{\psi}_{\text{MAP}}^*, \overline{\text{pred}}_{\text{MAP}})$ is $(\ell_{\text{MAP}}, \mathcal{P}_{\text{reinforce}})$-calibrated. $\qquad\square$

The proof of Theorem 4 makes use of the following technical lemma:

**Lemma 8.** *Let $\mathbf{p} \in \mathcal{P}_{\text{reinforce}}$. Let the sets $\Pi^*(\mathbf{p})$ and $\Pi(\mathbf{p})$ be defined as in the proof of Theorem 4 above. Then $\Pi(\mathbf{p}) \subseteq \Pi^*(\mathbf{p})$.*

*Proof of Lemma 8.* As in the proof of Theorem 4, for brevity, we will write $\ell_{\text{MAP}}$ as $\ell$ below.

We first observe that all permutations $\sigma \in \Pi(\mathbf{p})$ have the same value of $\mathbf{p}^\top \boldsymbol{\ell}_\sigma$. To see this, note that permutations in $\Pi(\mathbf{p})$ differ only in positions they assign to elements $i, j \in [r]$ with $u_{ii}^{\mathbf{P}} = u_{jj}^{\mathbf{P}}$. But since $\mathbf{p} \in \mathcal{P}_{\text{reinforce}}$, we have that if $u_{ii}^{\mathbf{P}} = u_{jj}^{\mathbf{P}}$, then $u_{i\gamma}^{\mathbf{P}} = u_{j\gamma}^{\mathbf{P}}$ for all $\gamma \in [r] \setminus \{i, j\}$. Thus, from the form of $\mathbf{p}^\top \boldsymbol{\ell}_\sigma$, we can see that if $u_{ii}^{\mathbf{P}} = u_{jj}^{\mathbf{P}}$, then interchanging the positions of $i$ and $j$ in a permutation $\sigma$ does not change the value of $\mathbf{p}^\top \boldsymbol{\ell}_\sigma$. This establishes that all permutations $\sigma \in \Pi(\mathbf{p})$ have the same value of $\mathbf{p}^\top \boldsymbol{\ell}_\sigma$.

We will show below that $\exists$ a permutation $\sigma^* \in \Pi(\mathbf{p}) \cap \Pi^*(\mathbf{p})$. This will give that $\sigma^* \in \Pi(\mathbf{p})$ and $\mathbf{p}^\top \boldsymbol{\ell}_{\sigma^*} = \operatorname{argmin}_\sigma \mathbf{p}^\top \boldsymbol{\ell}_\sigma$; by the above observation, we will then have that $\mathbf{p}^\top \boldsymbol{\ell}_{\sigma'} = \operatorname{argmin}_\sigma \mathbf{p}^\top \boldsymbol{\ell}_\sigma$ for *all* $\sigma' \in \Pi(\mathbf{p})$, i.e. that $\Pi(\mathbf{p}) \subseteq \Pi^*(\mathbf{p})$.

In order to show the existence of a permutation $\sigma^* \in \Pi(\mathbf{p}) \cap \Pi^*(\mathbf{p})$, we will start with an arbitrary element $\sigma^0 \in \Pi^*(\mathbf{p})$, and will construct a sequence of permutations $\sigma^1, \sigma^2, \ldots, \sigma^M = \sigma^*$ by transposing one adjacent pair at a time, such that all elements in the sequence remain in $\Pi^*(\mathbf{p})$, and the final permutation $\sigma^M$ is also in $\Pi(\mathbf{p})$.

Let $\sigma^0 \in \Pi^*(\mathbf{p})$. If $\sigma^0 \in \Pi(\mathbf{p})$, we are done, so let us assume $\sigma^0 \notin \Pi(\mathbf{p})$. Thus there must exist an adjacent pair of elements in $\sigma$ that are not ordered according to the scores $u_{ii}^{\mathbf{P}}$, i.e. there must exist $a, b, c \in [r]$ such that

$$\sigma^0(a) = c, \quad \sigma^0(b) = c + 1, \quad \text{and} \quad u_{aa}^{\mathbf{P}} < u_{bb}^{\mathbf{P}}.$$

Define $\sigma^1$ to be such that $\sigma^1(a) = c + 1, \sigma^1(b) = c$, and $\sigma^1(i) = \sigma^0(i)$ for all other $i \in [r]$. We will show that $\sigma^1 \in \Pi^*(\mathbf{p})$. For convenience let us denote $(\sigma^0)^{-1}$ as $\pi^0$ and $(\sigma^1)^{-1}$ as $\pi^1$. Note that

$$\pi^0(c) = \pi^1(c+1) = a$$
$$\pi^0(c+1) = \pi^1(c) = b$$
$$\pi^0(i) = \pi^1(i) \ \forall i \in [r] \setminus \{c, c+1\}.$$

From the expression for $\mathbf{p}^\top \boldsymbol{\ell}_\sigma$ in Eq. (2) in the proof of Theorem 4 above, we have

$$
\begin{aligned}
\mathbf{p}^\top \boldsymbol{\ell}_{\sigma^0} - \mathbf{p}^\top \boldsymbol{\ell}_{\sigma^1} &= \frac{1}{c} \left( \sum_{j=1}^{c} (u_{\pi^1(c)\pi^1(j)}^{\mathbf{P}} - u_{\pi^0(c)\pi^0(j)}^{\mathbf{P}}) \right) + \frac{1}{c+1} \left( \sum_{j=1}^{c+1} (u_{\pi^1(c+1)\pi^1(j)}^{\mathbf{P}} - u_{\pi^0(c+1)\pi^0(j)}^{\mathbf{P}}) \right) \\
&= \frac{1}{c} \left( \sum_{j=1}^{c} (u_{b\pi^1(j)}^{\mathbf{P}} - u_{a\pi^0(j)}^{\mathbf{P}}) \right) + \frac{1}{c+1} \left( \sum_{j=1}^{c+1} (u_{a\pi^1(j)}^{\mathbf{P}} - u_{b\pi^0(j)}^{\mathbf{P}}) \right) \\
&= \left( \frac{1}{c} - \frac{1}{c+1} \right) \sum_{j=1}^{c-1} (u_{b\pi^1(j)}^{\mathbf{P}} - u_{a\pi^1(j)}^{\mathbf{P}}) + \frac{1}{c}(u_{bb}^{\mathbf{P}} - u_{aa}^{\mathbf{P}}) + \frac{1}{c+1}(u_{ab}^{\mathbf{P}} + u_{aa}^{\mathbf{P}} - u_{ba}^{\mathbf{P}} - u_{bb}^{\mathbf{P}}) \\
&= \left( \frac{1}{c} - \frac{1}{c+1} \right) \left( \sum_{j=1}^{c-1} (u_{b\pi^1(j)}^{\mathbf{P}} - u_{a\pi^1(j)}^{\mathbf{P}}) + u_{bb}^{\mathbf{P}} - u_{aa}^{\mathbf{P}} \right) \\
&= \left( \frac{1}{c} - \frac{1}{c+1} \right) \left( u_{bb}^{\mathbf{P}} - \left( u_{aa}^{\mathbf{P}} + \sum_{j=1}^{c-1} (u_{a\pi^1(j)}^{\mathbf{P}} - u_{b\pi^1(j)}^{\mathbf{P}}) \right) \right) \\
&\geq \left( \frac{1}{c} - \frac{1}{c+1} \right) \left( u_{bb}^{\mathbf{P}} - \left( u_{aa}^{\mathbf{P}} + \sum_{j \in [r], j \notin \{c, c+1\}} (u_{a\pi^1(j)}^{\mathbf{P}} - u_{b\pi^1(j)}^{\mathbf{P}})_+ \right) \right) \\
&\geq 0,
\end{aligned}
$$

where the last inequality follows since $\mathbf{p} \in \mathcal{P}_{\text{reinforce}}$. This gives $\sigma^1 \in \Pi^*(\mathbf{p})$. Moreover, the number of adjacent pairs in $\sigma^1$ that disagree with the ordering according to $u_{ii}^{\mathbf{P}}$ is one less than that in $\sigma^0$. Since there can be at most $\binom{r}{2}$ such pairs in $\sigma^0$ to start with, by repeating the above process, we will eventually end up with a permutation $\sigma^M \in \Pi(\mathbf{p}) \cap \Pi^*(\mathbf{p})$ (with $M \leq \binom{r}{2}$). The claim follows. $\square$

**Proof of Theorem 6**

*Proof.* Let $\mathbf{p} \in \mathcal{P}_{\mathbf{f}}$. Define $\mathbf{u}^{\mathbf{P}} \in \mathbb{R}^r$ as

$$\mathbf{u}^{\mathbf{P}} = \mathbf{E}_{Y \sim \mathbf{p}}[\mathbf{f}(Y)] = \sum_{\mathbf{y} \in \mathcal{Y}} p_{\mathbf{y}} \mathbf{f}(\mathbf{y}) \, .$$

It is easy to see that $\mathbf{u}^{\mathbf{P}}$ is the unique minimizer of $\mathbf{p}^{\top} \boldsymbol{\psi}_{\mathbf{f}}(\mathbf{u})$ over $\mathbf{u} \in \mathbb{R}^r$.

Also define $\mathbf{y}^{\mathbf{P}} \in \mathbb{R}^{r(r-1)}$ as

$$y_{ij}^{\mathbf{P}} = \mathbf{E}_{Y \sim \mathbf{p}}[Y_{ij}] = \sum_{\mathbf{y} \in \mathcal{Y}} p_{\mathbf{y}} y_{ij} \quad \forall i \neq j \, .$$

For brevity, we will write $\ell_{\text{PD}}$ as $\ell$ below. Define the following sets:

$$\Pi^*(\mathbf{p}) = \operatorname{argmin}_{\sigma \in S_r} \mathbf{p}^{\top} \boldsymbol{\ell}_{\sigma} = \operatorname{argmin}_{\sigma \in S_r} \sum_{i=1}^{r} \sum_{j=1}^{i-1} (y_{ij}^{\mathbf{P}} - y_{ji}^{\mathbf{P}}) \cdot \mathbf{1}(\sigma(i) > \sigma(j))$$

$$\Pi(\mathbf{p}) = \left\{ \sigma \in S_r : u_i^{\mathbf{P}} > u_j^{\mathbf{P}} \implies \sigma(i) < \sigma(j) \right\} \, .$$

We claim that $\Pi(\mathbf{p}) \subseteq \Pi^*(\mathbf{p})$. To see this, let $\sigma \in \Pi(\mathbf{p})$. Since $\mathbf{p} \in \mathcal{P}_{\mathbf{f}}$, we have

$$y_{ij}^{\mathbf{P}} > y_{ji}^{\mathbf{P}} \implies u_i^{\mathbf{P}} > u_j^{\mathbf{P}} \implies \sigma(i) < \sigma(j) \, ,$$

$$y_{ij}^{\mathbf{P}} < y_{ji}^{\mathbf{P}} \implies u_i^{\mathbf{P}} < u_j^{\mathbf{P}} \implies \sigma(i) > \sigma(j) \, .$$

This clearly gives $\sigma \in \Pi^*(\mathbf{p})$. Thus $\Pi(\mathbf{p}) \subseteq \Pi^*(\mathbf{p})$.

By the definition of pred and $\Pi(\mathbf{p})$, we also have that $\exists \epsilon > 0$ such that for any $\mathbf{u} \in \mathbb{R}^r$,

$$\|\mathbf{u} - \mathbf{u}^{\mathbf{P}}\| < \epsilon \implies \operatorname{pred}(\mathbf{u}) \in \Pi(\mathbf{p}) \, .$$

Thus, we have

$$\inf_{\mathbf{u} \in \mathbb{R}^r : \operatorname{pred}(\mathbf{u}) \notin \operatorname{argmin}_{\sigma} \mathbf{p}^{\top} \boldsymbol{\ell}_{\sigma}} \mathbf{p}^{\top} \boldsymbol{\psi}_{\mathbf{f}}(\mathbf{u}) = \inf_{\mathbf{u} \in \mathbb{R}^r : \operatorname{pred}(\mathbf{u}) \notin \Pi^*(\mathbf{p})} \mathbf{p}^{\top} \boldsymbol{\psi}_{\mathbf{f}}(\mathbf{u})$$

$$\geq \inf_{\mathbf{u} \in \mathbb{R}^r : \operatorname{pred}(\mathbf{u}) \notin \Pi(\mathbf{p})} \mathbf{p}^{\top} \boldsymbol{\psi}_{\mathbf{f}}(\mathbf{u})$$

$$\geq \inf_{\mathbf{u} \in \mathbb{R}^r : \|\mathbf{u} - \mathbf{u}^{\mathbf{P}}\| \geq \epsilon} \mathbf{p}^{\top} \boldsymbol{\psi}_{\mathbf{f}}(\mathbf{u})$$

$$> \inf_{\mathbf{u} \in \mathbb{R}^r} \mathbf{p}^{\top} \boldsymbol{\psi}_{\mathbf{f}}(\mathbf{u}) \, ,$$

where the last inequality follows since $\mathbf{p}^{\top} \boldsymbol{\psi}_{\mathbf{f}}(\mathbf{u})$ is a strictly convex function of $\mathbf{u}$ and $\mathbf{u}^{\mathbf{P}}$ is its unique minimizer.

Since the above holds for all $\mathbf{p} \in \mathcal{P}_{\mathbf{f}}$, we have that $(\boldsymbol{\psi}_{\mathbf{f}}, \operatorname{pred})$ is $(\ell_{\text{PD}}, \mathcal{P}_{\mathbf{f}})$-calibrated. $\qquad\square$

**Proof of Theorem 7**

*Proof.* Let $\mathbf{p} \in \mathcal{P}_{\text{DAG}}$. Define $\mathbf{u}^{\mathbf{P}} \in \mathbb{R}^{r(r-1)}$ as

$$\mathbf{u}^{\mathbf{P}} = \mathbf{E}_{Y \sim \mathbf{p}}[Y_{ij}] = \sum_{\mathbf{y} \in \mathcal{Y}} p_{\mathbf{y}} y_{ij} \, .$$

It is easy to see that $\mathbf{u}^{\mathbf{P}}$ is the unique minimizer of $\mathbf{p}^{\top} \boldsymbol{\psi}_{\text{PD}}^*(\mathbf{u})$ over $\mathbf{u} \in \mathbb{R}^{r(r-1)}$.

For brevity, we will write $\ell_{\text{PD}}$ as $\ell$ below. Define the following sets:

$$\Pi^*(\mathbf{p}) = \operatorname{argmin}_{\sigma \in S_r} \mathbf{p}^{\top} \boldsymbol{\ell}_{\sigma} = \operatorname{argmin}_{\sigma \in S_r} \sum_{i=1}^{r} \sum_{j=1}^{i-1} (u_{ij}^{\mathbf{P}} - u_{ji}^{\mathbf{P}}) \cdot \mathbf{1}(\sigma(i) > \sigma(j))$$

$$\Pi(\mathbf{p}) = \left\{ \sigma \in S_r : \sigma \text{ corresponds to a topological order that could be returned by } \overline{\operatorname{pred}}_{\text{PD}}(\mathbf{u}^{\mathbf{P}}) \right\} \, .$$

We claim that $\Pi(\mathbf{p}) \subseteq \Pi^*(\mathbf{p})$. To see this, let $\sigma \in \Pi(\mathbf{p})$. Since $\mathbf{p} \in \mathcal{P}_{\mathrm{DAG}}$, we have that the graph with edge weights $(u_{ij}^{\mathbf{P}} - u_{ji}^{\mathbf{P}})_+$ formed by $\overline{\mathrm{pred}}(\mathbf{u}^{\mathbf{P}})$ is a DAG, and therefore $\sigma$ must agree with the edges in this graph, i.e.

$$
\begin{aligned}
u_{ij}^{\mathbf{P}} > u_{ji}^{\mathbf{P}} &\implies \sigma(i) < \sigma(j)\,, \\
u_{ij}^{\mathbf{P}} < u_{ji}^{\mathbf{P}} &\implies \sigma(i) > \sigma(j)\,.
\end{aligned}
$$

This clearly gives $\sigma \in \Pi^*(\mathbf{p})$. Thus $\Pi(\mathbf{p}) \subseteq \Pi^*(\mathbf{p})$.

Now, let

$$
A(\mathbf{p}) = \left\{ \mathbf{u} \in \mathbb{R}^{r(r-1)} : \overline{\mathrm{pred}}_{\mathrm{PD}}(\mathbf{u}) \notin \mathrm{argmin}_\sigma \mathbf{p}^\top \boldsymbol{\ell}_\sigma \right\} = \left\{ \mathbf{u} \in \mathbb{R}^{r(r-1)} : \overline{\mathrm{pred}}_{\mathrm{PD}}(\mathbf{u}) \notin \Pi^*(\mathbf{p}) \right\}\,.
$$

In order to show that

$$
\inf_{\mathbf{u} \in A(\mathbf{p})} \mathbf{p}^\top \boldsymbol{\psi}_{\mathrm{PD}}^*(\mathbf{u}) \quad > \quad \inf_{\mathbf{u} \in \mathbb{R}^r} \mathbf{p}^\top \boldsymbol{\psi}_{\mathrm{PD}}^*(\mathbf{u})\,,
$$

we will show that any sequence $\{\mathbf{u}_m\}$ in $\mathbb{R}^{r(r-1)}$ converging to $\mathbf{u}^{\mathbf{P}}$ must eventually lie outside $A(\mathbf{p})$, i.e. that any such sequence must eventually have $\overline{\mathrm{pred}}_{\mathrm{PD}}(\mathbf{u}_m) \in \Pi^*(\mathbf{p})$; the result will then follow by strict convexity of the function $\mathbf{u} \mapsto \mathbf{p}^\top \boldsymbol{\psi}_{\mathrm{PD}}^*(\mathbf{u})$ and the fact that $\mathbf{u}^{\mathbf{P}}$ is its unique minimizer.

Let $\{\mathbf{u}_m\}$ be any sequence in $\mathbb{R}^{r(r-1)}$ converging to $\mathbf{u}^{\mathbf{P}}$. Let

$$
\epsilon = \min_{i \neq j} \left\{ u_{ij}^{\mathbf{P}} - u_{ji}^{\mathbf{P}} : u_{ij}^{\mathbf{P}} - u_{ji}^{\mathbf{P}} > 0 \right\}\,.
$$

Then for large enough $m$, we must have the following (by convergence of $\{\mathbf{u}_m\}$ to $\mathbf{u}^{\mathbf{P}}$):

$$
\begin{aligned}
u_{ij}^{\mathbf{P}} - u_{ji}^{\mathbf{P}} > 0 &\implies u_{ij}^m - u_{ji}^m \geq \epsilon/2\,, \\
u_{ij}^{\mathbf{P}} - u_{ji}^{\mathbf{P}} = 0 &\implies u_{ij}^m - u_{ji}^m \leq \epsilon/4\,.
\end{aligned}
$$

Thus, for large enough $m$, the directed graph induced by $\mathbf{u}_m$ contains the DAG induced by $\mathbf{u}^{\mathbf{P}}$, and any edge $(i,j)$ such that $\mathbf{u}_{ij}^{\mathbf{P}} - \mathbf{u}_{ji}^{\mathbf{P}} > 0$ will not be deleted by the algorithm when $\overline{\mathrm{pred}}_{\mathrm{PD}}(\mathbf{u}_m)$ is evaluated. Thus, for large enough $m$, we have $\overline{\mathrm{pred}}_{\mathrm{PD}}(\mathbf{u}_m) \in \Pi(\mathbf{p}) \subseteq \Pi^*(\mathbf{p})$.

Since the above holds for all $\mathbf{p} \in \mathcal{P}_{\mathrm{DAG}}$, we have that $(\boldsymbol{\psi}_{\mathrm{PD}}^*, \overline{\mathrm{pred}}_{\mathrm{PD}})$ is $(\ell_{\mathrm{PD}}, \mathcal{P}_{\mathrm{DAG}})$-calibrated. $\qquad \square$