[Reviews · NeurIPS 2013]

Submitted by Assigned_Reviewer_6

The paper demonstrates calibrated convex surrogate losses for multiclass classification. If the n by k loss matrix (where the label set has size n and the prediction set has size k) has rank d, minimizing this surrogate loss corresponds to a d-dimensional least squares problem, and converting the solution to a prediction corresponds to a d-dimensional linear maximization over a set of size k. The paper describes two examples (precision at q and expected rank utility) for which n and k are exponentially large, but predictions can be calculated efficiently. It describes two more examples (mean average precision and pairwise disagreement) with exponential n and k for which the calibrated predictions apparently cannot be calculated efficiently, but efficiently computable predictions are calibrated for restricted sets of probability distributions.

This is an exciting contribution that seems to be important for a wide variety of multiclass losses of great practical significance. Technically, it is a combination of two results from ref [16] - the important low rank observation appeared already in [16]. The paper is clearly written.

The main criticism is of the results on calibration with respect to restricted sets of probability distributions. For instance, Theorem 4 gives a result that the efficiently computable prediction rule is calibrated for a certain family P_reinforce. Why is this family interesting? What is the intuition behind it? Are there interesting examples of distributions that satisfy these conditions? Similar questions apply to the set of distributions considered in Theorem 7.

Minor comments:
line 227: 1(\sigma^{-1}(i)\leq q) should be 1(\sigma(i)\le q). (Similarly at line 236)
242: 1 and 0 interchanged.
254: Having max(y_i-v,0) in place of simply y_i seems silly. Why not just redefine the set of labels as {0,1,...,s-v}^r?
320: u^p_{ij} is only defined for i\geq j. The appendix mentions that it is made symmetric, but not the paper.
Summary: The paper demonstrates calibrated convex surrogate losses for multiclass classification that are especially important when the loss matrix has low rank. This is an important contribution that applies to several significant losses.

Submitted by Assigned_Reviewer_8

Shows how to design a convex least squares style surrogate loss for high arity multiclass losses that have a low rank structure. Paper is very well written.

Only comment: given your introduction of the loss matrix \mathbf{L} why not state the condition in theorem 3 in matrix form?
Summary: This is a very nicely written paper that shows how to design convex surrogates for multiclass losses in manner that makes the surrogate more tractable. This works when the target loss has a low rank structure. As is explained in the paper a number of popular losses have this structure and the authors show how to construct the efficient surrogate.

Submitted by Assigned_Reviewer_9

The paper considers convex calibrated surrogates for consisitency. In particular, it constructs a least-square surrogate loss which can be calibrated for ranking problems associated with low-rank target loss matrix. The results seem novel and interesting. The results potentially are interesting to the NIPS audience since ranking is a popular topic.

However, the motivation on why you consider low-rank target loss matrix could be better motivated.
Summary: The paper addresses convex calibrated surrogates for ranking with low-rank loss matrix and the results seem novel and interesting. However, the motivation for considering low-rank loss matrix could be better motivated.

My review confidence is not certain since I have no research experience on ranking.
Author Feedback

Author rebuttal: We thank all the reviewers for the careful reading and helpful feedback. Below are brief responses to the main questions/concerns.

Reviewer 6 -

Intuition for noise conditions: The noise conditions in Theorems 4, 6, and 7 can essentially be viewed as conditions under which it is `clear and easy' to predict a ranking:

- Theorem 4: The ideal predictor (line 296) uses the entire $u$ matrix, but the predictor in theorem 4 uses only the diagonal elements. The noise condition P_reinforce can be viewed as requiring that the diagonal elements dominate and enforce a clear ordering among themselves. E.g. one simple (though extreme) example where the noise condition holds is when the relevance judgements associated with different documents (conditioned on the query) are independent of each other.

- Theorem 6: The noise condition P_f here requires that the expected value of the function f used in the surrogate must decide the `right' ordering. As mentioned in the paper, it is a generalization of the condition in Duchi et al. [11].

- Theorem 7: The condition P_DAG here is somewhat easier to understand than the others. In particular, it implies that the expected pairwise preferences are acyclic, and this is exploited by the algorithm used in the predictor. What makes this condition especially interesting is that it contains all the conditions in Theorem 6 (and therefore is more general than these conditions, including in particular that of Duchi et al. [11]).

We will try to include some of these intuitions for the conditions in the final version. Thanks also for pointing out the typos - we really appreciate it.

Reviewer 9 -

Motivation for considering low rank losses: The motivation is simply that such losses occur very often in natural problems. Indeed, all four ranking losses considered in the paper constitute such examples. Further examples can be found in various structured prediction problems, e.g. the Hamming loss used in sequence prediction constitutes another such loss.